# Attitudes and Practices Related to COVID-19 Vaccination with the Second Booster Dose among Members of Athens Medical Association: Results from a Cross-Sectional Study

**DOI:** 10.3390/vaccines11091480

**Published:** 2023-09-12

**Authors:** Georgios Zoumpoulis, Paraskevi Deligiorgi, Dimitrios Lamprinos, Panagiotis Georgakopoulos, Evangelos Oikonomou, Gerasimos Siasos, Georgios Rachiotis, Christos Damaskos, Dimitrios Papagiannis, Kostas A. Papavassiliou, George Patoulis, Fotios Patsourakos, Vasiliki Benetou, Elena Riza, Philippos Orfanos, Pagona Lagiou, Georgios Marinos

**Affiliations:** 1Emergency Care Department, Laiko General Hospital, 11527 Athens, Greece; gzoumpoulis@yahoo.gr (G.Z.); dimitrislamprinos@gmail.com (D.L.); panos.k.georgakopoulos@gmail.com (P.G.); 2First Department of Cardiology, Hippokration General Hospital, Medical School, National and Kapodistrian University of Athens, 11527 Athens, Greece; boikono@gmail.com (E.O.); gsiasos@med.uoa.gr (G.S.); 3Third Department of Cardiology, Thoracic Diseases General Hospital Sotiria, Medical School, National and Kapodistrian University of Athens, 11527 Athens, Greece; 4Department of Hygiene and Epidemiology, Faculty of Medicine, University of Thessaly, 41500 Larissa, Greece; grachiotis@gmail.com; 5Renal Transplantation Unit, Laiko General Hospital, 11527 Athens, Greece; x_damaskos@yahoo.gr; 6N.S. Christeas Laboratory of Experimental Surgery and Surgical Research, Medical School, National and Kapodistrian University of Athens, 11527 Athens, Greece; 7Public Health & Vaccines Laboratory, Department of Nursing, School of Health Science, University of Thessaly, 38221 Volos, Greece; dpapajon@gmail.com; 8First Department of Respiratory Medicine, “Sotiria” Hospital, Medical School, National and Kapodistrian University of Athens, 11527 Athens, Greece; konpapav@med.uoa.gr; 9Athens’s Medical Association, 11527 Athens, Greece; gipattt@gmail.com (G.P.); fnpatsourakos@gmail.com (F.P.); 10Department of Hygiene, Epidemiology and Medical Statistics, School of Medicine, National and Kapodistrian University of Athens, 11527 Athens, Greece; vbenetou@med.uoa.gr (V.B.); eriza@med.uoa.gr (E.R.); phorfanos@med.uoa.gr (P.O.); pdlagiou@med.uoa.gr (P.L.); gmarino@med.uoa.gr (G.M.)

**Keywords:** COVID-19, vaccine, physicians, vaccine hesitancy, booster vaccine

## Abstract

Background: There are limited data on the attitudes and acceptance of the second booster (fourth dose) of the COVID-19 vaccination among physicians. Methods: A cross-sectional, questionnaire-based, online study was conducted among members of the Athens Medical Association (A.M.A.) who were invited to participate anonymously over the period from January to March 2023. Results: From the 1224 members who participated in the survey, 53.9% did not receive the fourth dose of the COVID-19 vaccine. The main reasons for no vaccination were the lack of obligation to receive the fourth dose, the history of three doses of the COVID-19 vaccine and the lack of sufficient information about the effectiveness of the fourth dose. Over half of the three-dose-vaccinated participants were willing to receive the fourth dose in the near future. Interestingly, the vaccination coverage among participants who had been informed about the fourth dose through scientific sources was low. Conclusions: The low vaccination coverage with the fourth dose reported in this study can lead to broad and serious consequences, such as increase in COVID-19 infections, reduction of available healthcare staff and increased caseloads of COVID-19 in hospitals. Furthermore, hesitant physicians will adversely influence the vaccination uptake among the general population due to their key role in informing and recommending the vaccine. The healthcare system administration should acknowledge and address physician’s concerns through effective communication and better support.

## 1. Introduction

The sudden emergence of the coronavirus disease, named the COVID-19 disease, caused by SARS-CoV-2, was initially reported to the World Health Organization (WHO) by Chinese authorities in late December 2019 [1]. This novel disease has now become a worldwide concern, affecting citizens all over the world. Governments globally implemented strict measures, including the closure of air, maritime and land borders, in an attempt to isolate and control the spread of the disease. Despite these efforts, the global spread of the disease persisted, leading the WHO to declare COVID-19 as a pandemic on 11 March 2020) [1]. COVID-19 is not confined to specific age groups, genders or races. Nonetheless, certain underlying health conditions are considered risk factors and are associated with higher mortality rates. While individuals of all ages can get infected by the virus, middle-aged and elderly adults face a greater risk of hospitalization. Children on the other hand, have been found to be less frequently infected with COVID-19 and tend to exhibit milder symptoms when infected. Scientists and policy makers have had to face the rapid spread of the virus, its mutations and its ability to evade host defense mechanisms [2]. As an unpredictable event, and comparable to the economic scene of World War Two, the COVID-19 pandemic has had a deleterious effect on global healthcare systems with a ruffle effect on every aspect of human life as we knew it. To reduce COVID-19 cases, governments enforced quarantine measures in many countries [3]. Furthermore, the COVID-19 pandemic has also triggered psychological and physical burnout of healthcare personnel [4,5]. Various studies tried to provide explanations regarding severity, outcome and treatment of COVID-19 [6,7,8]. The global increase in COVID-19 cases created an urgent need for the development of safe and effective vaccines against the disease. However, even though several effective vaccines have become available, vaccine hesitancy in the general population decreased the possibilities of ending the pandemic [9]. Many countries such as France and Greece prioritized COVID-19 vaccination for health care workers (HCW’s) and people with comorbidities, who are all at high risk for severe coronavirus disease 2019 [10]. In Greece, the official national recommendations were two doses of mRNA vaccines administered at an interval of 8 weeks for Pfizer’s (Pfizer BNT162b2), Moderna’s (mRNA-1273) or AstraZeneca’s vaccines (ChAdOx1-S) [10,11]. Administration of a third dose was made obligatory at least 3 months after the basic vaccination with two doses for HCW’s and the general population because of its effectiveness on higher antibody titers compared to the first two doses [12]. The third dose could be any mRNA vaccine [11]. Complete vaccination was made mandatory for all HCW’s because of their high risk of infection with SARS-CoV-2 and the potential risk of virus transmission among themselves, their patients and their relatives [10]. As they have the main role of treating COVID-19 and administering vaccinations, healthcare workers are delegated to influence vaccine uptake. According to a cross-sectional survey about the willingness of healthcare workers in Switzerland, if HCWs are themselves hesitant, it is important to first promote vaccine uptake in this specific population [13]. The most frequent reasons for vaccine hesitancy among HCWs include concerns about vaccine safety and side effects, the perception that using personal protective equipment is sufficient and that COVID-19 is not threatening to them [13]. According to a previous study conducted in the same population in Greece during the early stages of the pandemic in 2021, the acceptance of the mandatory vaccination was almost 90% with the hesitancy reasons to be pending the vaccination appointment and fear of side effects [14]. The National Vaccination Committee of Greece in the updated tables of the national vaccination program for the year 2023, for the general population including HCWs, did not include the COVID-19 vaccination [15]. Given the ongoing discussions regarding the safety and effectiveness of COVID-19 vaccines, understanding the pivotal role of public health communication in fostering greater acceptance of these vaccines has become increasingly essential [16,17,18]. Currently, the vaccines used are bivalent, which include two different strains of the virus to provide better protection against COVID-19’s latest variants [19]. In Greece, the National Vaccination Committee recommends updated bivalent COVID-19 vaccines for use as a single booster dose at least three months following the primary vaccination, or booster vaccination or after COVID-19 infection, to specific vulnerable population groups (e.g., people 60 years old and older, healthcare workers, people 12–59 years old with underlying diseases and caregivers of people with underlying diseases) [20]. The aim of this study was to assess the attitudes and perceptions of physicians, members of the Athens Medical Association, regarding the fourth dose of COVID-19 vaccination and find potential deviations from the previous study in 2021.

## 2. Materials and Methods

Our study was based on an anonymous online survey, distributed to the members of Athens Medical Association (A.M.A.). The data were collected over the period from January to March 2023. All members of A.M.A. were invited to complete the questionnaire. Inclusion criteria were active membership with A.M.A, access to the internet and the voluntary involvement of physicians located in Athens. The study followed the principles of the Declaration of Helsinki as revised in 2008. The participants provided informed consent before the questionnaire’s completion. The protocol of the study was approved by the Board of the A.M.A. (Code: 12.01.2023).

To estimate the number of responders for this study we used the formula for descriptive studies provided from OpenEpi, for confidence levels of 99.9%.

A total of 1224 A.M.A. members participated in the study (response rate 5%). 

The questionnaire was based on the previous questionnaire used to evaluate the coverage of COVID-19 vaccination and associated factors among physicians, enriched with new questions to assess the attitudes towards the booster dose. The survey included questions on socio-demographics (sex, age and occupational characteristics) and close-ended questions regarding perceptions about the importance, safety and effectiveness of vaccines and opinions regarding side effects. In addition, the questionnaire included close-ended questions on the COVID-19 vaccination history (“Have you been vaccinated against COVID-19 with the fourth dose”, answer options: Yes/No; “Are you willing to do the fourth dose?”, answer options: Yes/No), and influenza vaccination coverage for flu season 2022–2023: “Have you been vaccinated with the influenza vaccine (season 2022–2023)?” (Yes/No). In the case of no vaccination against COVID-19, the participants were requested to report the reason for non-vaccination with the fourth dose (predefined options: non-obligation of the fourth dose; history of 3 doses of the vaccine; insufficient information for the fourth dose). Information about the general perception for receiving a COVID-19 vaccine were also obtained (“Which of the following factors most influenced your view of the COVID-19 vaccine?”, answer options: safety concerns; I am not at risk of COVID-19 disease; the time of the development of the vaccines was short; I am using homeopathy drugs; pending vaccination appointment; and I am opposed to vaccinations). Physicians were asked to assess using a 4-point Likert scale the significance of vaccinations for Public Health, the efficacy and safety of vaccines. They were also prompted to express any potential concerns regarding possible side effects. Furthermore, data concerning their source of information about COVID-19 booster doses were collected, as well as asking them to evaluate the reliability of information from the Greek Public Health Authorities regarding COVID-19 vaccination with the fourth dose. Last, information about the perceptions of the physicians on the obligation of the first doses of COVID-19 vaccine were collected (“Do you agree with the initial mandatory vaccination schedule against COVID-19 for medical personnel to protect public health?”, answer options: Yes/No). The questionnaire used can be found in Appendix A. 

Categorical variables were displayed in terms of both absolute counts and relative frequencies (%), while age was represented as mean ± standard deviation. To find statistically significant differences among categorical variables, the chi-square test (χ^2^) was employed, and for age, the *t*-test was used. To explore the factors associated with the acceptance of a fourth dose of vaccination, a multiple logistic regression analysis was conducted. Statistical significance was determined with a *p*-value below 0.05. The analysis was carried out using SPSS 23 for Windows.

## 3. Results

Table 1 presents the demographics of the study sample. A total of 54.9% (n = 672) of the study participants were males, 44.9% (n = 550) were females and 0.2% (n = 2) answered gender-neutral. The mean age was 53.5 years (SD = 10.87, range 24–83 years). Among the responding physicians, 64.1% (n = 785) were working in the private sector, 27.9% (n = 341) in the National Health System (N.H.S.), 4.3% (n = 53) were working with the Greek Army and 3.7% (n = 45) were working in university hospitals. Nearly three out of four (n = 903, 73.8%) of the responders agreed with the mandatory vaccination for the medical personnel regarding the initial COVID-19 vaccination scheme. Slightly over half of the physicians (52.5%, n = 642) did not take the fourth dose of the COVID-19 vaccine with the main reasons being the lack of obligation to receive the fourth dose (53.9%, n = 346), the history of three doses of COVID-19 vaccine (36%, n = 231) and the insufficient information about the effectiveness of the fourth dose, (10.1%, n = 65) (Table 2). More than half of the 642 three-times-vaccinated physicians (52.3%, n = 336) were willing to receive the fourth dose soon. In Table 2, the general reasons of the three-times-vaccinated physicians’ decision to not accept the fourth dose are presented. The main reason was that their appointment was still pending (32.7%, n = 210), followed by the perceptions that the time of the development of the vaccines was short (22.9%, n = 147) and that they were not in danger of COVID-19 (22.6%, n = 145). Almost 1/5 of the responders were afraid of the side effects (10.5%, n = 128). Other reasons mentioned by a minority were the opposition to vaccinations (0.6%, n = 4) and the receipt of homeopathy medication (1.2%, n = 8). 

Table 3 presents the characteristics and perceptions of study participants overall and separately for those who did and did not receive the second booster. Physicians who perceived that vaccines are generally safe, effective and useful for the protection of public health were less willing to receive the fourth dose of the COVID-19 vaccine. On the contrary, older age, the perception that information received from the Greek Public Health Authorities regarding COVID-19 vaccination with the fourth dose was reliable and the lack of fear of vaccine-related side effects were significantly associated with COVID-19 vaccination uptake. History of influenza vaccination for the flu season 2022–2023 was associated with higher vaccination coverage with the second booster (fourth) dose against COVID-19. Moreover, physicians who were being informed about COVID-19 vaccines through reliable scientific sources (scientific journals, CDC, ECDC, WHO, website of the National Health System and website of Athens Medical Association) recorded lower vaccination coverage with the fourth dose against COVID-19. We did not find significant difference regarding the fourth dose of COVID-19 vaccine by sex or type of employment (e.g., private/public sector). Multiple-regression derived odds ratios (ORs) assessing the association of several variables with vaccination with the fourth dose are presented in Table 4. Non-fear over the side effects of the fourth dose of the COVID-19 vaccine (OR = 2.22, 95% C.I. = 1.68–2.94), history of influenza vaccination for the 2022–2023 season (OR = 17.34, 95% C.I. = 10.89–27.63), age above 53 years old (OR = 1.49, 95% C.I. = 1.15–1.93) and perception that the information on COVID-19 vaccination from the national public health authorities was reliable (OR = 2.35, 95% C.I. = 1.75–3.16) were found to be independently associated with an increased probability of vaccination against COVID-19 with the fourth dose of the vaccine. The perceptions of physicians regarding safety, effectiveness and importance for public health, and scientific publications as a source of information for the booster doses, were not associated with the fourth dose. 

## 4. Discussion

This is a cross-sectional study evaluating the perceptions as well as the coverage of the booster COVID-19 vaccine among members of the largest medical association in Greece over a period of two months in 2023. A substantial proportion of the physicians who responded to the questionnaire, more than half, were not vaccinated with the booster dose of COVID-19 vaccine because, as they reported, this dose was not obligatory, they were already vaccinated with three doses of the COVID-19 vaccine and had insufficient information about the effectiveness of the fourth (booster) dose. The data shows us a reduction in the vaccine uptake, compared to the high rates (almost 90%) of the early pandemic [14]. 

The first country that approved the fourth dose of the COVID-19 mRNA vaccine was Israel [21]. A retrospective cohort study conducted in the Israeli population showed that older adults, over 60 years old, who had received a first booster dose at least 4 months before the second booster had lower mortality and hospitalization rates due to COVID-19 during the Omicron surge and were remarkably diminished for those who had received a second booster dose [22]. Although the Omicron variant appeared to cause milder symptoms than earlier variants [23], the unparalleled stream of SARS-CoV-2 infections drove the Israeli authorities to authorize the second booster vaccine dose to protect the most vulnerable patients from possible severe illness COVID-19. Nevertheless, the foresight of the second booster before an approval in the United States and Europe was extremely controversial [24,25]. The ECDC suggested that a second booster (fourth) dose could be provided to adults 80 years of age and above because they are at higher risk of severe COVID-19 and should be protected by a fourth dose [26]. Another retrospective cohort study conducted in Israel showed that the second booster (fourth) dose significantly increased the protection against manifestation of severe symptoms [27]. Also, a cross sectional study among HCWs in Mongolia showed higher support for mandatory COVID-19 vaccination (93.7%) compared to general vaccination (77.8%) [28]. On the other hand, a national online survey of 27 HCWs in Switzerland found that participants showed resistance to COVID-19 mandates and that such mandates would discourage them of working in the system [29]. Our deviation from the high rates of other countries regarding booster vaccination could be supported by the low number of COVID-19 positive cases in our country being, according to ECDC, between the five last countries with new positive COVID-19 cases (positivity rate below 4%), the complete abrogation of the restrictive measures and the return to normality, factors that could lead to the change of physicians’ practices towards vaccination.

Interestingly, more than 50% of the three-times-vaccinated physicians were willing to receive the fourth dose soon. One of the main reasons was the rejection of all COVID-19 vaccines in the past due to safety concerns; so, the recorded high rate of willingness for vaccination of our study could be explained as the time elapsed after the first doses is sufficient to persuade the hesitant population that the vaccines are safe. We also recorded the main reasons in general that influenced the COVID-19 vaccination acceptance, with most of the participants answering pending vaccination appointment, followed by the perception that the time of the development of the vaccines was short and that they are not in danger from COVID-19. Compared to the data of the previous study in Greece in 2021, we noticed an increase in the lack of fear of COVID-19 infection among the participants, findings that could be explained by the milder symptomatology of the current variants. The suggested second booster dose may have been conceived as evidence of the vaccine’s ineffectiveness by some [30]. On the other hand, an important predictor for booster dose acceptance was found to be its effectiveness against severe illness, symptomatic infection, and community transmission [31]. 

A very small percentage of the physicians in our sample were against vaccinations. There were 73.8% of the participants that agreed with the mandatory vaccination for the medical personnel regarding the initial COVID-19 vaccination scheme. Interestingly, in our survey, physicians who were being informed about COVID-19 vaccination through reliable scientific sources (scientific journals, CDC, ECDC, WHO, website of the National Health System and website of Athens Medical Association) recorded lower vaccination coverage with the fourth dose against COVID-19, data that contrast with our previous findings, where scientific sources were significantly associated with COVID-19 vaccine uptake [14]. We assume that the reason for that is the limited available evidence for the fourth dose’s effectiveness. Also, another reason could be the mild cases of COVID-19, which make healthcare workers feel safer and believe that they do not need the fourth dose of COVID-19 vaccine. As the largest percent of the physicians in our study had scientific publications, which according to recent data reported that the long-term protection of booster doses remains unclear, as their source of information, our findings could be a consequence of this input [31,32,33,34]. On the other hand, our study showed that the belief that information collected from the Greek Public Health Authorities concerning COVID-19 vaccination with the second booster (fourth dose) were reliable and being fearless over the possible side effects, were significantly associated with COVID-19 vaccination; findings that positively affect vaccine uptake since the early stages of the pandemic [14]. Lastly, older age was associated with increased likelihood of receiving the fourth dose of COVID-19 vaccine among physicians. This finding is in line with the results of our previous study [14] and may be related to the increased prevalence of comorbidities in physicians in the older age groups.

Another interesting finding is that the history of influenza vaccination for flu season 2022–2023 was associated with higher vaccination coverage with the fourth dose of the COVID-19 vaccine, data that verify the findings of the previous study in Greece [14]. The percentage of vaccinated participants against influenza for the 2022–2023 flu season (76.6%) is similar, even higher, compared to the 2020–2021 season [35], findings that show the positive attitude of physicians to vaccination, and the belief that vaccines against serious diseases like influenza are essential for the protection of Public Health. Similar to our results, an umbrella review conducted among healthcare workers and healthcare students, in low and lower middle-income countries, revealed that people who had received a vaccination in the past five years or were willing to accept an influenza shot were more likely to accept a COVID-19 vaccine [36,37]. According to a cross-sectional study, it has been shown that besides gender, marital status, history of prior influenza vaccination, the number of confirmed and suspected cases, vaccine efficacy and vaccination convenience, the recommendation given from doctors is an independent predictor for the general population COVID-19 vaccination uptake [38,39]. Also, it has been shown that people who have refused in the past to get vaccinated against other illnesses, like influenza, are more likely to be hesitant of COVID-19 vaccine uptake [30].

Almost 1/5 of the responders were afraid of the side effects (10.5%, n = 128). Based on an assessment of adverse effects subsequent to COVID-19 vaccination, a substantial proportion of individuals experienced adverse effects such as myocarditis, skin conditions and glomerular diseases. These conditions have been associated with mRNA vaccines; they are considered infrequent occurrences [40]. People who experienced adverse effects from previous vaccine doses were presenting them as a reason for accepting or rejecting the fourth dose [18,27,29].

We did not find significant differences regarding the fourth dose of COVID-19 vaccine by gender and type of employment of the physicians who participated in our survey. Our findings are consistent with our previous study in Greece [14]. Nevertheless, a study showed that gender was associated with acceptance of the fourth dose, presenting a higher level of reluctance among women regarding the uptake of the fourth dose of the vaccine. These results align with findings from other studies that demonstrate variation in vaccination uptake based on gender [21]. 

Several limitations apply to our findings, which need to be considered during interpretation. Notably, the regional sample that was used does not fully represent the entire national population of the physicians. Nonetheless, it is our contention that the data presented herein satisfactorily capture the intentions of Greek physicians concerning the uptake of a booster dose of COVID-19 vaccine. This assertion is based on the inclusion of participants from one of the most prominent medical associations in Greece. Secondly, we were not able to make a causal inference due to the cross-sectional study design. Thirdly, our study was questionnaire-based and there is a potential for information bias. Moreover, we were not able to obtain responses from all invited participants and this may be a source of selection bias. Furthermore, our study’s questionnaire was distributed to all the physicians working in Athens, including those who were working in hospitals with direct involvement with COVID-19 patients and those who did not treat COVID-19 patients. That fact could influence their decision about vaccination with the fourth dose, affecting the results of our study. Also, the lack of data regarding characteristics of the physicians such as type of specialization, type of employment (intern, resident, fellow and attending physician) and length of service did not give us the opportunity to find other possible factors affecting the acceptance of the vaccine. Additionally, the mean age of our study population presupposes increased probability of chronic medical conditions of the participants that could influence their acceptance of the booster dose. Lastly, because of the large number of vaccinations done, HCWs are more likely to meet patients presenting with serious side effects, which reflexively provokes unconscious bias regarding their beliefs about the safety of vaccines [40,41].

## 5. Conclusions

All concerns and beliefs that have been expressed by physicians concerning the second booster (fourth) dose of the COVID-19 vaccine emphasize that HCWs have an executive role. On the one hand, physicians form part of the healthcare system and on the other hand, they are part of the general population who have their personal lives, their personal beliefs and values, which can affect their attitudes and hesitancy towards disease prevention interventions. The low vaccination coverage with the fourth dose among physicians can lead to broad consequences. Physicians have an executive role in COVID-19 vaccination programs, as they are the people who are most trusted by the general population for making decisions to get vaccinated. In conclusion, hesitant physicians may influence lower intervention uptake among the general population. The healthcare system should acknowledge their concerns, provide support and address them through effective communication.

## Figures and Tables

**Table 1 vaccines-11-01480-t001:** Demographics of the participating physicians.

		N	%
	Male	672	54.9
Gender	Female	550	44.9
	Gender-neutral	2	0.2
	Total	1224	100
	Physicians working in NHS	341	27.9
	Physicians working in the private sector	785	64.1
Type of employment	Physicians working in universities	45	3.7
	Physicians working with the Greek Army	53	4.3
	Total	1224	100
Age (years, mean, SD)		53.5 ± 10.87	

**Table 2 vaccines-11-01480-t002:** Reasons affecting the acceptance of vaccination with COVID-19 vaccine and the fourth dose of the COVID-19 vaccine.

Reasons to not accept the 4th dose	N	%
Non-obligation of the 4th dose	346	53.9
History of 3 doses of the vaccine	231	36.0
Insufficient information for the 4th dose	65	10.1
Total	642	100
**Reasons to not accept COVID-19 vaccine**		
Pending vaccination appointment	210	32.8
The time of the development of the vaccines was short	147	22.9
I am not at risk of COVID-19 disease	145	22.6
Fear of side effects	128	19.9
Receiving homeopathy medication	8	1.2
I am opposed to vaccinations	4	0.6
Total	642	100

**Table 3 vaccines-11-01480-t003:** Characteristics and perceptions of study participants overall and by vaccination coverage with the second booster.

Variable	COVID-19 Vaccination Coverage with the 4th Dose		
	Yes (%)	No (%)	*p*-Value *
N	582	642	
Sex			
Male	319 (47.5)	353 (52.5)	
Female	263 (47.8)	287 (52.2)	0.400
Neutral	0	2 (100)	
Age (Years, Mean, SD)	54.5 ± (11.4)	52.5 (10.3)	0.002
Employment status			
Working in the public sector (NHS, Army, Universities)	196 (44.6)	243 (55.4)	0.128
Working in the private sector	386 (49.2)	399 (50.8)	
The vaccines are important for Public Health			
Fully Agree/Agree	572 (48.6)	606 (51.4)	<0.001
Fully disagree/Disagree	10 (21.7)	36 (78.3)	
In general, vaccines are safe.			
Fully Agree/Agree	574 (49.0)	598 (51.0)	<0.001
Fully disagree/Disagree	8 (15.4)	44 (84.6)	
In general, vaccines are effective.			
Fully Agree/Agree	566 (49.0)	590 (51.0)	<0.001
Fully disagree/Disagree	16 (23.5)	52 (76.5)	
I am concerned over vaccination side effects.			
Fully Agree/Agree	200 (39.1)	312 (60.9)	<0.001
Fully disagree/Disagree	382 (53.7)	330 (46.3)	
Information received from the Greek Public Health Authorities regarding COVID-19 vaccination with the 4th dose was reliable.			
Fully Agree/Agree	419 (53.8)	360 (46.2)	<0.001
Fully disagree/Disagree	163 (36.6)	282 (63.4)	
Are you vaccinated against flu for 2022–2023?			
Yes	560 (59.8)	377 (40.2)	<0.001
No	22 (7.7)	265 (92.3)	
The information regarding the 4th dose of COVID-19 vaccine was from			
Scientific sources	503 (49.7)	509 (50.3)	0.001
Social media and independent websites	79 (37.3)	133 (62.7)	

* Chi-square test (χ^2^) was used for examining differences in proportions except for age where the *t*-test was applied.

**Table 4 vaccines-11-01480-t004:** Odds ratios (ORs) and 95% confidence intervals (95% CI) derived from the multiple logistic regression assessing the association of several factors with the uptake of the 4th dose of the vaccine.

Independent Variable	OR	95% C.I.	*p*-Value
Age			
53 and above	1.49	1.15–1.93	0.003
Below 53	1.00 (ref)		
Reliable information from Greek Public Health Authorities			
Yes	2.35	1.75–3.16	<0.001
No	1.00 (ref)		
Fear of COVID-19 4th dose of vaccine side effects			
No	2.22	1.68–2.94	<0.001
Yes	1.00 (ref)		
Influenza vaccination for flu season 2022–2023			
Yes	17.34	10.89–27.63	<0.001
No	1.00 (ref)		
In general, vaccines are safe			
Yes	2.02	0.33–12.45	0.45
No	1.00 (ref)		
The vaccines are important for public health			
Yes	0.594	0.83–4.27	0.60
No	1.00 (ref)		
In general, vaccines are effective			
Yes	1.47	0.49–4.41	0.49
No	1.00 (ref)		
Source of information			
Scientific sources	1.20	0.842–1.72	0.35
Social media and independent websites	1.00 (ref)		

## Data Availability

The study data are available from the corresponding author on reasonable request.

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
