# Peer review of "Attitudes and Practices Related to COVID-19 Vaccination with the Second Booster Dose among Members of Athens Medical Association: Results from a Cross-Sectional Study"

_vaccines, 2023, doi:10.3390/vaccines11091480_

Round 1

Reviewer 1 Report

The manuscript (ID: vaccines-2566609) aimed to assess the attitudes and perceptions of physicians, members of the Athens Medical Association regarding the 4th dose of COVID-19 vaccination.  

Comments (Major revision):     

  • In the work as a whole, check and correct the use of `,` and `.` in numbers. 
  • Lines 119-125: List the inclusion and exclusion criteria of subjects in this study. Explain whether it is only working physicians. Are retired doctors included in the survey?
    • Note: Table 2 shows `The mean age was 53,5 years (SD=10,87, range 24-83 years).`.  
  • Line 238: Explain why the variable `Age` did not enter the multivariate logistic regression model shown in Table 4?   
  • Lines 319-320: Explain why the information about the variable `Age', which is stated and discussed in this sentence, is not shown on Table 4?  
  • Lines 347-353: Clarify the role of gender in this study. Was gender statistically significantly associated or not statistically significantly associated with acceptance of 4th dose of COVID-19 vaccine among Greek doctors in this study?     
  • Lines 354-364: In the subsection Limitations of this study, it is necessary to discuss the influence of the study design (that is, a cross-sectional study design), the absence of data on other characteristics of doctors (type of specialization, length of service, etc.), whether the doctors included in this by the survey, were employed in jobs where they were in direct contact with patients with COVID-19 (during their treatment and care) or maybe they worked in other jobs without direct involvement in the treatment of patients with COVID-19 (e.g. scientific work, or teaching in schools), as well as potential influence of the medical condition of the doctors in terms of chronic diseases and other comorbidities on the acceptance of the 4th dose of the COVID-19 vaccine.  
  • Lines 365-378: It is not a good practice to list references in the Conclusions section. Correct this, so that only the most important results presented in the current study are highlighted in the Conclusions section.  

The quality of English language is appropriate. 

Author Response

Dear reviewer,

We are grateful for the valuable suggestions and your feedback.

Please, see the word file where we answer your comments point by point.

Reviewer 2 Report

The manuscript entitled” Attitudes and Practices Related to COVID-19 vaccination with the second booster dose among members of Athens Medical Association: results from a cross-sectional study “ nicely fall under the special issue: COVID-19 Vaccine Acceptance and Uptake: Insights from Behavioural and Social Sciences. Here the authors did a cross sectional study to detect the attitudes and practices of members of Athens Medical association who received 2nd COVID 19 vaccine. This is a nice study. Lots of data were collected and analyzed systematically using robust and appropriate statistical analysis. In addition, the authors also have describe the limitations, too may though!

There are many studies focusing the KAP of people of different class/groups etc towards covid 19 vaccines. This study is not something new, but considering the location and target population it carries some new findings.

Comments:

Please add the  questions use in to get the data as suppl. material.

In methodology, provide more information if the questions were open or closed??

Line 126. The  results shows that only 5% people responded among the A.M.A members participated. Can it be explained why the response rate was so low??

Line 287. Provide more data on what was the number of covid19 positive case in your country and those in other countries to confirm that the rate was really low in Greece then other countries.

Line 290: use number 50 instead of fifty please

Line 336 to 339, no need, please delete this already focused earlier..

Line 371, 4th not fourth, be unform.

Why not try to do a PCA analysis of these variable to see which are more closer related variable influencing the Attitudes and Practices

Author Response

(The authors gave the same response as above.)

Round 2

Reviewer 1 Report

Thank you for the opportunity to re-review manuscript ID:  vaccines-2566609. 

Comments (Minor revision):      

  • The Authors have answered some of the comments, but not all.  
  • Also, the explanations provided by the Authors do not satisfactorily follow the corrections entered in the revised version of this manuscript.  
  • Regarding point 1, I assume that the commas and full-stops in the numbers will be corrected during the editing since the Authors stated that they made the corrections which in fact were not made.   
  • Also, the description of Table 4 is not appropriate and is incorrect in certain parts. First of all, the Authors had to see that the results for the variable "Age" which they entered in the revised version of the paper are in fact statistically significant (the Authors state that "In our test models we had found that factor age didn't affect the results"). Check and correct this in an appropriate manner in the text of the Results.  
  • Also, the Authors should check the values for OR, CI and also the p value both throughout the paper and the Tables and align them.  

The quality of English language is appropriate. 
